# Tough Love—Impactful, Caring Coaching in Psychologically Unsafe Environments

**DOI:** 10.3390/sports10060083

**Published:** 2022-05-25

**Authors:** Jamie Taylor, Michael Ashford, Dave Collins

**Affiliations:** 1Grey Matters Performance Ltd., Stratford upon Avon CV37 9TQ, UK; mike@greymattersuk.com (M.A.); dave@greymattersuk.com (D.C.); 2Faculty of Science and Health, School of Health and Human Performance, Dublin City University, Glasnevin, D09 NA55 Dublin, Ireland; 3Moray House School of Education and Sport, The University of Edinburgh, Holyrood Campus, Edinburgh EH8 8AQ, UK; 4Faculty of Health and Life Sciences, Coventry University, Coventry CV1 5FB, UK

**Keywords:** psychological safety, care, elite performance, talent development, challenge

## Abstract

(1) Background: The interpersonal dimensions of coaching in high performance sport have been subject to increasing scrutiny but with limited evidence to guide practice. Similarly, there is increasing practical interest in the concept of psychological safety, often portrayed as an implicitly desirable characteristic of all sporting environments but, as yet, still to receive research attention in high performance. As a first step to addressing these deficiencies, the present study addressed two research aims: (a) to examine the extent to which matched groups of international and released professional rugby union players perceived psychological safety to be an adaptive feature of their developmental experience and (b) to understand what elements of the player’s coaching experience were perceived to be enabling or disenabling of future progress. (2) Methods: Seven rugby union players who had ‘made it’ and eight players who had been released from their professional contracts took part in a semi-structured interview exploring their developmental experiences. Data were subsequently analysed using Reflexive Thematic Analysis. (3) Results: Both groups of players found each of their talent development and high performance environments to be psychologically unsafe. Furthermore, players perceived coaches who were the most impactful in their development as offering ‘tough love’. This included a range of ‘harder’ and ‘softer’ interpersonal approaches that presented the player with clear direction, role clarity and a sense of care. It appeared that this interpersonal approach helped the player to navigate, and benefit from, the psychologically unsafe high performance milieu. (4) Conclusions: There appear to be a number of balances for the coach in the high performance setting to navigate and a need for more nuance in applying constructs such as psychological safety.

## 1. Introduction

In recent times there has been a significant interest in the development of talent towards high performance (HP) sport [1]. This has generated a significant volume of publications from across contexts [2] and enormous sums of money spent across the world to promote athlete development. Despite this interest, however, there remains a paucity of research that is ‘for’ the coach, rather than ‘of’ coaching [3,4]. In essence, there is limited research that offers the ‘granularity’ [5] necessary for coaches and practitioners to make truly evidence-informed decisions [6]. This may be one of the reasons why research appears not to have made significant impact on applied practice, where a wide variety of different, even contradictory, implicitly and explicitly held constructs influence coaching [7]. Specifically in HP sport, there is a limited body of literature that has investigated truly ‘elite’ performers [8]. Similarly, there has been a failure to distinguish between levels of performance and understanding further development when athletes do reach the elite level [9].

### 1.1. Psychological Safety

In the applied domain, there is growing interest in a construct with a long history in the organisational literature, psychological safety (PS) [10]. Although there are multiple definitions in use, the concept was initially defined by Edmondson [11] as: ‘a shared belief by members of a team that the team is safe for interpersonal risk taking’, underpinned by a number of scale items, including: the lack of ‘fear of mistakes being held against you’, ‘it is safe to take a risk on this team’ and ‘working with members of this team, my unique skills and talents are valued and utilised’ [11]. In the most recent definition, Edmondson suggests a dual effect: ‘in psychologically safe environments, people believe that if they make a mistake or ask for help, others will not react badly’ [12]. Perhaps in recognition that there are issues with the application of Edmondson’s definition in sport, a recent paper redefined PS as: ‘the perception that one is protected from, or unlikely to be at risk of, psychological harm in sport’, with harm including perceptions of threat or fear [13].

Several issues with the application of the construct in HP have been raised. As highlighted in a recent critical review, even by redefining the construct, it is difficult to see how it can be applied to athletes in HP [14]. At the very least, it has been suggested that there is a need to apply PS in a critical manner, understanding potential downsides such as lower overall performance without accountability, or even unethical behaviour [15,16]. Similarly, there is a need to recognise the realities of the HP milieu, one in which there will nearly always be consequences for poor performance [17,18] that may challenge one’s implicit safety.

Notably, despite a wave of interest in the applied domain, there remains very limited empirical investigation of the construct in sport. For instance, current examples involve high level university team sport athletes [19] and amateur rugby [20]. There is even less at higher levels of performance, for example NCAA basketball [21] and, notably, given the complex and multidimensional nature of the construct, no qualitative data that we are aware of.

### 1.2. Challenging Coaching

This lack of empirical investigation is notable, as one of the most robust findings in talent development (TD) research is the central role of challenge dynamics in promoting elite performance [22,23,24,25,26]. An example of these dynamics is the impact of early advantage in talent pathways, with athletes experiencing less challenge earlier in the pathway being more likely to be deselected as the challenge level increases [27,28]. As a result, it has been proposed that athletes can benefit from a range of affective conditions through development, as they experience a range of positive and negative experiences, so long as they can deploy the requisite psycho-behavioural skills to learn from their experiences [29,30]. Therefore, in the HP domain, it appears that a number of situations have the potential to be particularly emotionally disturbing and therefore a useful means to provoke further development [30]. Given that in nearly all cases, the coach will be responsible for a significant number of emotionally disturbing athlete experiences, whether they be positive or negative (e.g., selection), the coach has a central role to play in the wider development of the athlete, which may usefully provoke perceptions of an unsafe environment.

A number of conceptual framings of coaching practice have been suggested to understand the characteristics of effective TD environments, or ‘all aspects of the coaching situation’ [31]. For the coach seeking to take advantage of these dynamics, several features of TD have the potential to be impactful for the athlete including coherence of experience [32], individualisation [33] and the quality of messages received and sent by the athlete [34]. Yet, despite understanding what characterises effective coaching practice, limited literature has actively considered what the coach actually does. For example, the work of Collins and colleagues [23] suggested that the TD coaching of ‘super champion’ athletes was characterised by a blend of high challenge, facilitative and relatively non-directive input, in contrast to less successful athletes where the coach drove the agenda.

There is a similar limited volume of empirical work considering the role of the HP coach, particularly as regards the enhancement of performance. As an example, the work of Lara-Bercial and Mallett [35] found the need for coaches to enforce a culture of high expectations and standards, producing an inherently challenging environment. They also introduced the term ‘driven benevolence’ to describe an approach where coaches cared for others and maintained optimal development through pressure. As such, there appear to be a number of interpersonal dimensions that are particularly pertinent to the coaches of current or future high performers. These include the need for setting high expectations, ensuring that athletes have role clarity [36] and doing this with a level of emotional ‘elasticity’ to make and communicate tough decisions, albeit in a considerate manner [35].

### 1.3. Care

Elsewhere there has been a growth in the literature examining the role of care in coaching. Whilst seen as a core feature of coaching practice, it is often portrayed as an addition to performance or development [37]. In contrast, the recent literature has drawn on the work of Noddings’ theory of care in pedagogical relationships [38,39] to suggest a dual strand of care, where the coach should care ‘for’ and ‘about’ the athlete. To ‘care for’ an athlete, coaches express ‘devotion and desire’ to meet the needs and wants of that individual [40,41]. Such devotion is characterised by a coach’s engrossment, motivational displacement and reciprocity in the coaching relationship. Engrossment refers to the sustained attention of the coach in relation to the athlete’s needs and wants, consistently offering high levels of empathetic concern. Motivational displacement captures the behaviour of a coach who aims to meet the athlete’s needs even if it supersedes that of their own. Finally, reciprocity implies that the care offered to a player can only be confirmed when the player receives and acknowledges it. In contrast, caring about an athlete describes a coach’s emotional concern for an individual, without the need for engrossment, motivational displacement nor a reciprocal interaction.

Interestingly, others have highlighted limitations to this gentle view [41,42], suggesting that the nature of the HP milieu provides individuals with a social context where coaches may care for athletes through interactions that are harder in nature. For instance, Cronin and colleagues [40] explored the role of care within an English Premiership football environment, where the milieu was characterised as cut-throat, competitive and volatile [43,44]. They offer the conclusion that care is inherently contextual and situated within a social milieu [40]. It is this view that sits in contrast to views of dichotomous perspectives of coaching practice, particularly in HP, that have suggested a separation of person and performer, with an emphasis on the former. This false dichotomy encourages the view that care in coaching is more ‘caring about’, than ‘caring for’ an athlete. If the latter is about meeting the wants and needs of the individual, particularly in HP populations, it is likely that it will be expressed differently to a participant with different motivations [45]. Worthy of note is the idea that high performers, or ambitious developing athletes, may want or require a more robust coaching approach, which is already established in the literature [30]. Yet, confusingly, it has recently been suggested that the concept of ‘tough love’ may be used as a euphemism for abusive coaching practice [46]. This seems at odds with the suggestion that care should be understood within particular social circumstances; that those aspiring to an elite level of performance may require a different approach from their coaches, and the often-used idea that coaching requires holding people to high standards. Reflecting these apparent conflicts, there is a need to understand what athletes perceive to be truly developmental coaching.

As a result of the dynamics presented in the literature and the lack of overall empirical investigation, as practicing coaches and coach developers working in TD, we felt a need to investigate the dynamics involved in development and the role of the coach in this process. Thus, we sought to understand the experiences and perceptions of athletes who had progressed through a TD pathway, reaching or failing to achieve senior elite status. Given the need to understand what was considered effective and ineffective, two matched groups of successful and unsuccessful athletes were considered appropriate as a means of understanding different experiences. This takes account of calls for comparison groups in talent research [2] and the value of using negative case studies, rather than risking survivorship bias in successful samples [47]. Our ultimate purpose being to inform the practice of coaches working with athletes in the TD or HP setting. As a result, the aims of this study were: (a) to examine the extent to which two matched groups of international and released professional rugby union players perceived PS to be an adaptive feature of their developmental experience and (b) to understand what elements of the player’s coaching experience were perceived to be enabling or disenabling of future progress.

## 2. Methods

### 2.1. Research Philosophy, Design and Methods

Given the aims of this study and our wish to critically explore the practical utility of popular ideas and concepts in HP sport, a pragmatic research philosophy was employed [48]. The pragmatic approach encourages the use of methods that provide findings and implications that are practically meaningful, without need to adhere to a specific epistemological view [49]. Thus, the comparison of player’s perceptions of PS and their coaching experiences, between those who ‘made it’ and those who were released provides a pertinent investigation for anyone interested in the elite sport context.

In line with our pragmatic approach, qualitative research methods allow for a deep examination of the player group’s developmental experiences [50]. Qualitative research methods allow authors to collect rich descriptive data, with an aim of producing a useful interpretation of a practical problem, rather than one that is absolute [51]. Furthermore, a pragmatic research philosophy encourages the consideration of biases and preferences to make sense of findings. Reflecting these considerations, it is important to note that this study was aided by our experience as active practitioners within HP sport and specifically in rugby union [52,53].

### 2.2. Participants

Two purposefully sampled groups of male rugby union players were recruited against distinct inclusion and exclusion criteria. In both cohorts, players were matched by the criteria of: (a) progressing through the English Premiership academy system between 2012 and 2018, (b) had represented their country at junior international level at either U18 and/or U20 level and (c) had signed a professional contract at a senior elite team in the English Premiership (the highest level of performance nationally). As a point of difference, the first group were players whose career status was defined as ‘made it’. All had progressed through the domestic game and subsequently been selected to play for their country at senior international level (*n* = 7; Age, *M* = 22.14). Importantly, for perspective on the career status of the player, interviews took place within 6 months of players entering an international camp or playing for their country for the first time. The second group were players whose career status was defined as those who were ‘released’. Those who, despite matching the first cohort as having progressed through the academy system, played junior international rugby and signed a professional contract, had been released from their professional contract (*n* = 8; Age *M* = 22.75; See Table 1 for participant career status). The comparison between these two groups allowed us to explore the developmental experiences and perceptions of coaching along similar pathways that resulted in divergent outcomes. Finally, all players were recruited to take part through personal contact and, following the protocol approval by the University Ethics Committee, completed informed consent.

### 2.3. Data Collection

Both groups of players were invited to participate in two stages of data collection, both of which took place within a wider semi-structured interview. A semi-structured interview guide was developed and refined through a pilot interview with former professional players (*n* = 2) with similar profiles to the released population, minus experience of junior international rugby. These pilot studies led to subtle refinements of the semi-structured interview guide, including clearer guidance regarding the use of graphic timeline and the refinement of questions to enhance clarity. The interview consisted of open-ended questions that elicited responses informed by appropriate literature whilst follow-up probes and prompts were planned for and used to allow expansion on key points [54]. Interviews were conducted by both the first (*n* = 10) and second (*n* = 5) authors. Rather than for the purpose of data analysis and instead as a means to enhance recall and generate rich dialogue, the first stage of the interview asked players to sketch out their playing journey in rugby union on a timeline, highlighting key moments and transitions for the player along the *Y* axis [55]. Players were asked to score their relative development in terms of progress (+5) and stagnation (−5) along the *X* axis. Following this initial stage, the interview then asked players to go through the timeline again, this time focusing more deeply on specific perceived critical periods and, importantly, the time between perceived critical events. The second stage of the interview included questions regarding players’ perceptions of PS, as per definition of Edmondson [11] and coaching experiences within each age/stage of their pathway at academy club, junior international, senior club, and for the made it group, senior international levels. The interview guide is available on request from the first author. Abiding by regulations put in place by the University Ethics Committee to mitigate against the risk of COVID-19, interviews were arranged over video-conferencing software (Zoom Video Communications, California, Version 5.7) at a time and date that suited the participant. Prior to this, a pre-briefing allowed them to reflect on the timeline task and interview questions ahead of the interview. The interviews lasted between 60 and 105 min (*M* = 79 min) and were recorded for subsequent analysis.

### 2.4. Data Analysis

All interviews were transcribed verbatim and subsequently checked for accuracy against audio recordings. Data were then analysed using a Reflexive Thematic Analysis (TA) approach [56], using QSR NVivo Version 12 software. Against our pragmatic orientation, TA was chosen given the need to examine patterns of shared meaning across the player cohort [57]. Indeed, a core feature of TA is the recognition that the researcher plays a key role in the process of generating themes through engagement with the data. This allowed for deep reflexive engagement between researcher, data and theory (Braun and Clarke, 2019). Analysis was conducted by the first author, utilising each of the six phases initially outlined by Braun and Clarke [58]. Importantly, this took place flexibly, with appropriate non-linear movement between phases [56]. At the first stage, the first author became familiar with the content, highlighting and annotating areas of interest. Second, coding was conducted on a surface (semantic) level, before capturing the assumptions that underpin surface meaning through multiple sweeps of analysis [57]. Third, initial themes were generated, organised and captured from the initial coding process. At the fourth stage, the second author, acting as a critical friend, supported the review and refinement of themes to quality check if they were ‘coherent, consistent and distinctive’ [59]. The fifth phase involved defining and naming themes based on attribution of shared meaning. The final stage was the write up and report of data [57].

### 2.5. Trustworthiness

Several measures were taken to ensure trustworthiness in our approach. First, member reflections were solicited by email following completion of the six-phase TA process [60]. This involved all participants being contacted and sent a tabulated form of the final themes through the TA to seek their reflections on generated themes. In addition, participants were asked if the themes reflected their own experiences and if they had any further comments or considerations. Nearly all participants chose to take part in member reflections (*n* = 14 from 15) and their additional reflections have been incorporated into the results section. Players were universally supportive of the concept of ‘tough love’ (as defined by this study), and something they perceived as crucially important to their overall development. One way in which member reflections deepened overall analysis was the emphasis that players put on the technical and tactical competence of the coach, in addition to the interpersonal and pedagogic dimensions of their practice. As an example, player 9 commented: ‘as regards to accountability, I am all for that and with coaches who really care about my development. But it is so important that they hold you accountable for the right stuff, stuff that would have genuinely improved me’.

In addition, during data collection, the first author kept a reflexive journal reflecting on key differences and similarities between players’ perceptions of PS and coaching experiences and key areas of interest in line with the research questions. This journal was also used as an audit trail, to critically consider the methodological approach and support the initial generation of codes [61]. Finally, to ensure resonance in our approach, the second author, who is an experienced qualitative researcher, acted as a critical friend throughout the process [59].

## 3. Results

The purpose of the study was two-fold. Firstly, to understand the perceptions of both groups of players to understand the extent to which perceptions of PS were an effective and clear feature of their development experience. Secondly, to explore what elements of the player’s coaching experience were perceived to be enabling or disenabling of future progress. The first part of the results section addresses the first of these research questions, with two developed themes. The first developed theme concerns the extent to which player’s perceived it to be possible that the HP rugby union environment had the potential to offer them a sense of PS. The second generated theme presents the adaptive and maladaptive consequences of these perceptions.

### 3.1. A Lack of Psychological Safety

Across both groups of players, a lack of PS as a near constant factor in their professional career was reported. Players described this lack of safety as being driven by the judgement inherent to the HP milieu, making them subject to consistent scrutiny and criticism. Yet, for this group, the core feature of their experience that appeared to drive perceptions of a lack of safety was the judgement conferred by selection.

There’s no comfort, no safety, if you don’t work hard or you perform poorly, there’s eight other guys that are going to happily step in your shoes. It can be exhausting, but it definitely helps. It’s one of those places where you bring your gumshield to breakfast, you don’t want to put a foot wrong.(S6)

Players perceived their coaches to be central in engineering a lack of safety. They expressed that this was due to the coach’s role in making selection and contractual decisions:

It’s the most uncomfortable environment I’ve been in…if you were making a mistake because you weren’t switched on, it was a major issue. I wasn’t on it for a session, (coach) picked up on it straight after, he told me that I was going to be in the starting 15, but I’d worked my way out of it and that it would be tough to get back in. It was a good lesson for me, I reacted my usual way, when I’ve had a kick up the a*se, that’s where I’ve sat down and properly had a think about what I need to do and where I need to be…when I’m under that intense pressure, I feel a bit on edge, at the time it doesn’t feel great, afterwards, you know you’ve progressed a lot.(S5)

Additionally, where players were part of strong playing squads, this lack of safety was also experienced because of peer comparison, selection pressure and the consequent intra-group competition. In essence, at both the individual and organisational level, players felt that safety was incongruous with the nature of the HP milieu.

### 3.2. A Double-Edged Sword

Rather than these perceptions being debilitating, the ability to cope with and develop in a psychologically unsafe environment appeared critical to their progress. As a particularly prominent factor amongst the ‘made it’ group, players appeared to respond adaptively to a lack of safety. Take for example player three’s (S3) early experience of international rugby:

(International coach) pulled me in on the Wednesday morning before we trained. He says: ‘we’re gonna start you’…I woke up a day later to a text saying you need to see (coach). He said: ‘I’m not picking you, you didn’t train well, XXX had the upper hand on you’. I went away and had a think, I knew what (coach) wanted and what he was trying to do…I had to find gears I never had before in training. I thought I was leaving it all out there, but somehow, I wasn’t. It brought far more out of me. The game is mad, you can be on top of the world for 48 h before and then rock bottom. That camp was a perfect example and although it was traumatic at the time, I look back on it and I loved it, it definitely improved me.

As a comparison, consider the response of player eight (S8), reflecting on deselection from an international U20 squad:

A couple of players got picked ahead of me and I was thinking, what more could I be doing? I’m playing well, starting in (on loan in lower league) ahead of players who are on the bench (at a lower level). Why is this fair? Why is this happening? It especially annoyed me because I felt like they took (player) because of his brother. It was a stitch up.

As such, what appeared to be adaptive or positive was not the perception of safety. Rather, it was how individual players responded to the nature of the milieu. Indeed, maladaptive responses to this lack of safety were a prevalent and indeed, derailing factor for some of those players unable to transition from age group international to senior elite performance. One that seemed to be exacerbated where a player’s developmental experience had been overly safe and lacking the demands of the senior game: ‘through the academy, life was easy, rugby was easy. Then, you go from being big fish in a small pond to a small fish in a huge pond. I found it so hard’ (R6). Notably, players who perceived their development experience to be very safe, expressed that such environments did not adequately prepare them for the demands of the HP milieu.

This does not suggest that players believed that they were consistently able to respond adaptively; their experiences seemed to also change over time. Whilst the nature of the milieu remained the same, it was the player that seemed to change. Take for example the changing experiences of player four (S4) as he began to deploy psycho-behavioural skills to manage the demands placed on him as he reflected on two stages of his career. Firstly, as he began in the senior training environment:

In my first year, I was terrified of making mistakes. You’re new on the scene, you’re a young guy and thrown into training with players like XXX [a senior international and club stalwart] and you don’t want to f*ck up. I felt I trained within myself for a lot of that period. I’ve addressed that now but there’s still a lot of pressure…nobody’s position is safe.

Players from both groups held the perspective that a lack of safety, rather than being debilitating, was an adaptive feature of their environment. Note the perceived differences for player four (S4), as he began embedding himself into a senior international squad:

When you don’t have expectation, you don’t perform at the level you need to. If you feel like a bit safe, when there’s no expectation, you think you can coast. When everyone is competing for a spot, that gets the best out of everyone. The squad depth at (international) is ridiculous. Everyone’s battling so hard to play, it’s so competitive. I thrive in that. I like the intensity but there’s a couple of players who don’t thrive in those environments.

When players did perceive a level of safety, often reported to be outside of the HP milieu, this was seen subsequently as a negative factor in their development; seemingly prompting stagnation over the longer term. In the case of player 13 (R6), when he was loaned out to a semi-professional team:

It sounds silly, but I was very confident in my position (at loan club). I wasn’t going to dropped for a 19 year old student or a 28 year old plumber. I think that’s the problem, especially during that time. I needed to be pushed, I just wasn’t. I accepted that I’ll be playing for this team every weekend on loan.

It appeared that, rather than seeking safety and the certainty of their position, players were appreciative of being held accountable for their performance and the processes necessary for their continued development. Where this accountability and judgement was absent, it was perceived to be a negative developmental factor. Players also perceived this to be important at the group level, where the collective response to a lack of safety was seen as a key determinant of team performance:

(Club coach), when you don’t get selected, looks at how hard you train. It’s almost 10 times harder when you’re not getting picked because you got to push yourself even harder. In that sense it’s a good environment, when you play the non-selected in training and all of them are all p*ssed off, that preps you for the weekend. It’s probably a difference between teams because in some places players don’t get picked and sulk. This place is completely different, it fires people up.(S3)

Players also described a lack of safety as prompting increased effort and attention to detail. The pressure conferred by a lack of safety was also significantly fatiguing and long-term exposure left players exhausted:

I was just so tired, not so much physically, but mentally. I just couldn’t get away from all the pressure. The lockdown (COVID) came at just the right time for me, I really enjoyed it. It was just what I needed because those two years, not doing well at (club) were really tough. I took that quite personally and struggled with it. So the lockdown was almost like a restart button.(S1)

I was straight out of school. I finished the (international U18) stuff, finished my last exam then the next day I was straight into (club). I couldn’t get used to the pressure, needing to be on it all the time. I was tired and just didn’t cope with it, it was a chore from the beginning.(R7)

The perception of fatigue was most strongly felt where levels of safety were lowest and for extended periods of time. Despite this, however, there was a perception that this was highly developmental, both in the short term, driving them to higher standards and greater effort and, when this pressure was reduced, there was a perception that players were better able to cope with lower pressure in other environments:

Being at (international), you feel like you have to step up a level, or you will just get sent home. It feels like you are on X Factor! I feel like I have to be on point every session or I am going to be embarrassed. You have (coach) prowling around watching everything as well. Everything that you do is going to get judged, it either pushes you into your shell, or drives you to perform. I found it hard to adjust, but going back into club rugby, I felt so much more relaxed. I felt like I could go at the same intensity, but I wasn’t as stressed out, it was like I had adapted to the stress and could get the same results. It helped me massively from a mental point of view.(S7)

Players’ reflections were complex; there was a highly individual and contextually mediated response to a lack of PS. In nearly all cases, players felt that high pressure and a lack of safety was very uncomfortable at the time, but through the deployment of different psycho-behavioural skills, negative impacts were moderated and could become a developmental stimulus. Where players lacked the psycho-behavioural skillset to cope, or where the pressures were prolonged and uncontrollable, the inherent lack of safety was a significant risk factor for the player’s overall development.

### 3.3. Tough Love

In seeking to understand what approaches players considered to be the most effective in supporting their development, two themes were developed as prominent features of their experience. The first theme, coaches use of ‘harder’ approaches in their coaching practice, concerned holding players accountable to high standards, giving them role clarity, engaging in robust feedback processes and attention to detail. The second generated theme included the ‘softer’ approaches used by coaches in offering the player a level of openness and care. Perhaps summarising the needs that players perceived they had throughout the early stages of their careers, player three (S3) noted: ‘I’ve worked with so many good coaches, we always had good conversations. They know when someone needs an arm on the shoulder or know when they need a kick up the a*se’. The reflections of players were complex and multidimensional; players believed that they derived important but differential benefits from different coaching behaviours. This didn’t appear to be a balance between hard and soft approaches. Instead, players perceived coaches as offering more of one or the other through to offering both simultaneously. Across the interviews, both groups frequently used the term ‘tough love’ to describe what they perceived to be the most enabling coaching approach. Players consistently reflected on their desire and the necessity for the coach to adopt ‘hard’ approaches (tough) and the need for ‘softer’ approaches (love).

Table 2 presents the data based on player perception, identifying the approaches used by coaches that they perceived to be enabling of their future progress and the opposite pole [62], perceived to be disenabling. The perspectives of the players suggest a non-dichotomous perspective on effective coaching. Players strongly believed that effectiveness was the result of both ‘harder’ and ‘softer’ approaches to coaching practice. In short, players strongly desired highly competent coaches who presented them with a clear direction and held them to high standards, whilst also caring for and about them.

#### 3.3.1. Harder Approaches

Across both groups, there was a near universal perception that players wanted coaches who adopted ‘harder’ approaches to their coaching. Their descriptions indicated that harder approaches seemed to offer players direction, motivation and robust feedback, guiding the reflective patterns that appeared to be supportive of further performance development. For example, player 13 (R6) suggested:

I needed a coach, who offered some tough love, who would be on top of me, that would be honest, but also respect me. I only got that in the early years of my career. Other coaches, who I had a really good relationship with, there was no: ‘this wasn’t good enough’. I needed a coach to say: ‘your performance at this level is fine, but I want you to think about your ambitions as a player and if you want to push on, your performance at the weekend was crap’. There was never any of that. I knew if I went out with no preparation, had a beer the night before, hadn’t really done much mental prep…I’d still have been a seven out of ten. I needed coach to tell me: ‘seven out of ten at this level is not good enough’.

There was a strong perception across the group that the challenges of the elite game required these harder approaches to drive and promote the extra levels of performance necessary to make the jump to truly elite performance. Players also reflected on the need for role clarity to make sense of where they stood in a challenging environment and mitigate against the ill effects of a lack of PS.

It was a complete emotional rollercoaster, one week I was captain, the next week I couldn’t captain the tiddlywinks team. I had no idea where I stood, it was like the team performance rested on me. I heard from another coach that one week I was being recommended to (international coach), being offered 2 year contract on way more money, the next (club coach) wasn’t going to pick me at all. Whether there was an agenda to get rid of me I don’t know, but there was a lot going on behind the scenes.(S1)

The absence of role clarity was felt strongest in relation to the frequent experience of emotionally disturbing challenges, such as selection and contractual matters. This lack of clarity was experienced most strongly amongst the group of ‘didn’t’ players but was also a feature of the whole group’s experience. Without knowing where they stood within an environment, or what they needed to do in order to obtain opportunities to further their careers, players felt confused and directionless:

I was told that I’d hit all goals set for me, but the goals had changed. I got told I wasn’t going to be contracted, but that I’m doing exactly what they’d asked of me, but they’d moved the goal posts at some point. I hadn’t been told.(R8)

Players were also well aware of the scale of the challenge presented by the HP milieu and were accepting of the commitment necessary to reach the elite level. Coaches who guided this process were perceived to engage players in robust feedback processes, that were often emotionally laden but acted to give clear direction to the player and enabled them to take action:

(Head coach) was known to be a bit of a ‘yes man’. He’d always say: ‘yeah, yeah, you’ll get your chance’. The type of person who avoids the tricky conversations. Other coaches would just tell you something like (Head coach), you’d try and do it, go and do it but it wouldn’t change anything.(R4)

Where robust feedback was offered to the player, it was seen as a performance enhancer, helping players to make sense of the difficulties they faced. The opposite of which, players found especially difficult to understand or energise their next steps:

Wishy washy feedback was a killer. I was being released and it just wasn’t clean: ‘we really rate you, but we aren’t contracting you.’ I go away and what the hell am I supposed to do with that? Now, it’s like: ‘you’re not getting picked because of this, this and this.’ It’s just being straight up and honest. I think coaches need to think deeply about their approach to those conversations, they can kick players on, or beat them down.(S3)

Players also put significant weighting on the ability of the coach’s competence in guiding their progress. This was seen both in terms of the coach’s competence and knowledge base, but also through their individual rather than group-focused attention to detail:

When you dropped that ball, or when you made that pass, he asked: ‘what do you think you could have done?’. (Coach) remembers all the details, and it was a huge attention to detail. It just broke everything down for me, it was so much easier. Their attention to detail, it was eye opening.(S6)

Notably, where coaches did not have the perceived knowledge base, particularly as players reached higher levels of the game, regardless of other facets of support, they were rejected as helpful agents.

#### 3.3.2. Softer Approaches

Regarding the player’s perceptions of ‘softer’ approaches used by coaches, ‘softer’ approaches in the mind of the players did not seem to mean gentle. Instead, these reflected a genuine concern for their welfare and performance, offering them a voice to express thoughts and concerns:

(1st team coach) was tough in terms of having high expectations for everybody in the squad and to take responsibility. No one was able to sort of rest on their laurels, there was an expectation for everyone to improve, all the time. But also, this person was a generous, kind and compassionate human being…From my point of view, I thought it was absolutely fantastic.(R2)

This compassion was reflected by the attention paid to the player as an individual; understanding their broader life, with the perception that the player was able to approach the coach regarding their life outside of the game. However, this perception of care was also multidimensional, and, more prevalently, players referred to care as being a coach’s investment of time and effort in their development as a performer. In essence, it seemed that, whilst players valued coaches paying attention to and recognising their wider lives, players cared more about coaches who invested in their careers as athletes.

The final generated theme concerned the extent to which coaches made themselves open to conversations or invited a player’s input. Coaches who were perceived as open, allowed players to approach them to discuss issues and to understand the player’s point of view: ‘Having spent time with a lot of coaches, those that are open enough to invite challenge and debate is such a massive thing for my development’ (S7). For others, with coaches where these conversations did not take place, there seemed to be a barrier for coaches in understanding the psycho-emotional state of the player. In some cases, players felt that coaches were deliberately undermining them and had closed themselves off to the input of players. Where players perceived this, it seemed to change the player’s interpretation of the coach’s input. For example, in one case, a coaching group was perceived to favour a certain type of player based on non-performance relevant characteristics:

(International age group coaches) didn’t want me there, it was basically all private school lads down there and the coaches were quite cliquey because nearly all the players had been in their system for ages. I remember in my first session I made a tackle and I folded a (club) back rower. The coaches were like: ‘oh the (club) lads are here’. I was the only player from my club, it was clear that they didn’t like me. They had their favourites from certain clubs and the louder ones who were more extroverted. I couldn’t approach them.(R3)

In addition to the perceptions of the player regarding the extent to which the coach could offer a ‘tough love’ approach, it appeared that the key mediating factor was the extent to which the player’s perceived the coach to be competent. Often demonstrated through technical understanding of the sport, or of a player’s needs, where a coach’s competence was questioned, it often led to their input being rejected.

I respected coaches more if I thought they knew what they were talking about. With (international age group), I was told I wasn’t fit enough but I ran a 4.51 bronco (fitness test), it was so annoying, they didn’t know what I needed to work on. They were just covering their a*ses.(R5)

At times, players held different perspectives on the same behavioural approach from the same coach. Compare, for example, the perceptions of players one and seven of a 1st team coach:

I was like, wow, this man who’s achieved so much is willing to take a gamble on me. He was really direct with me, but I remember feeling like I had to give everything I can to learn from him.(S1)

For me, it is finding the coaches who are doing it for the right reason, and you look at their track record, you know they are doing something right. Some coaches will just shout at you for the sake of it, (coach) was just an authoritarian, I stopped listening.(S7)

In essence, amongst this group of players, there was a nuanced necessity for the coach to deploy a range of interpersonal skills as a medium to support their development. This was experienced on a highly individual basis and often moderated, not only by the type of relationship between coach and athlete, but also perceptions of competence, power and the wider network of relationships held by the coach with other players.

## 4. Discussion

The aims of this study were: (a) to examine the extent to which two matched groups of international and released professional rugby union players perceived PS to be an adaptive feature of their developmental experience and (b) to understand what elements of the player’s coaching experience were perceived to be enabling or disenabling of future progress. The findings present two overarching points of discussion including the player’s perceptions regarding PS, firstly the near universal lack of PS throughout their career and second, the role of the coach, including their experiences of interpersonal dimensions and soft skills of their coaches throughout the pathway. Subsequent evidence-informed implications are offered to coaches and practitioners within HP and TD settings.

### 4.1. Psychological Safety

In line with recent critiques of the uncritical application of PS as a conceptual model in HP sport, data in this study suggest a complex and nuanced relationship between perceptions of safety and performance development [14]. Much like data collected in other contexts, it does appear that the construct may not always be a performance enhancer [15]. The experiences of the players suggest that a lack of safety appeared to be a ubiquitous feature of their experience and, indeed, where players had the requisite psycho-behavioural skills to cope with a lack of safety, it appeared to be a performance enhancer [55]. Indeed, this may be a key differentiating feature of the HP milieu and non-elite settings, cf. [20]. In this regard, with growing and significant interest in the construct, there remains a lack of empirical investigation and limited conceptual clarity [13]. In essence, the players in this sample were well aware that if they made mistakes, it would be held against them, that risk taking was not safe and the pressures of selection meant that their unique talents would not always be valued [11,12]. In addition, the response to a lack of safety seemed to differentiate between those that were able to make the next step in the professional game towards senior international status and those who were not. Yet, importantly, the present sample also alluded to a number of potential issues associated with these perceptions of a lack of safety, including the high fatigue associated with the pressure to perform over extended periods [63].

Selection and deselection are clear realities of the HP milieu; in the case of elite rugby union, faced on a weekly basis. Coaches simply cannot offer athletes an environment where mistakes will not be held against you, where risk taking is safe and unique skills will always be valued [11]. Taking a different definition, nor is it possible to offer an experience that is: ‘protected from, or unlikely to be at risk of, psychological harm in sport (including fear, threat, and insecurity)’ with shared perceptions of comfort [13]. Therefore, in terms of theory, rather than aiming for PS as a universally desirable outcome, we might be a little more realistic and consider safety on a continuum, with differential effects depending on the extent of the lack of safety. Low levels of safety (when coupled with an appropriate psycho-behavioural skillset) potentially drive increased levels of effort and attention to detail, whilst higher levels of safety may allow the athlete the space to experiment and recover from the demands placed on them, cf. [29].

This would suggest that for the athlete, if there are differential effects from more or less safe experiences, through appropriate planning, it may be desirable to periodise the extent of the pressure experienced by the athlete, through an athlete’s career [33,64]. Either way, a more individual view of safety is likely appropriate [13], rather than the blanket suggestion that an entire environment can be classified as being ‘safe’ [65]. In essence, if the concept of PS is to offer a meaningful impact in HP sport, there is a need for further critical investigation, framed by a real world understanding of the HP milieu. In short, and as with so much else in coaching, ‘it depends’ on the coach using professional judgement and decision making (PJDM) to select the most appropriate approach for each context [66].

### 4.2. Role of the Coach

Coherent with the findings of previous research, athletes put a significant weighting on the interpersonal dimensions of their interactions with coaches [67]. Importantly, the players highlighted multiple coaches, along with other stakeholders, playing an active role in their development [34,68]. The interpersonal dimensions of these interactions were complex and in many ways seemed to contradict a variety of coach education advice [23]. Players did not perceive the role of the coach to be offering unremitting positivity. Instead, players sought genuinely developmental input from their coach [29]. This input required the coach to be knowledgeable across multiple domains [69] and operationalise this through a variety of interpersonal approaches [70,71].

Prominently, the coach’s ability to help the player generate appropriate role clarity was supportive of development and protected against maladaptive consequences of a lack of PS. That is, whilst players were well aware of the judgement and the inherent potential for them to be deselected, understanding the criteria they would be judged on and help from the coach to make progress towards these ends appeared to be central to the coach’s role. For the most part, the coach appeared to enable this by deploying a variety of ‘harder’ approaches. Importantly, however, the longer-term vehicle for improvement was also supported by ‘softer’ approaches [72]. This appeared especially important where players perceived a significant performance gap between themselves and the senior elite level.

#### 4.2.1. Interpersonal Dimensions

This has significant implications for the existing care literature in coaching. Rather than ‘tough love’ being used as a euphemism to justify abusive coaching practices, ‘tough love’ was perceived by players to be an essential feature of coaching practice that characterised both the hard and soft approaches that they deployed [46]. The data here present an orthogonal relationship between the role of the coach and players’ perceptions of their decisions and interactions. Coaches who were perceived by players to offer them ‘tough love’ were characterised as both caring for and caring about their players [39,40]. Our data are consistent with the assertions that a consideration of ‘care’, needs to be ecologically situated [40] and take account of the nature of the HP milieu. Put simply, context is key. Similar to research conducted in TDEs [42], it also suggests that care does not always need to be demonstrated through softer approaches. If we take this view in relation to the findings within this study, players offer numerous accounts where care is expressed by their coaches through interactions that, without reference to social context, may be perceived as uncaring [73,74]. These include instances of stern corrective feedback, the threat of deselection, increasing psychological and social pressures and the impact of contractual decisions. It seems unanimous from the findings that the players perceived effective coaching interactions as tough but caring, as they were often perceived as necessary given the challenges they regularly face in their day-to-day life as a professional rugby union player [42,75]. Additionally, players were clear that overly positive interactions with coaches, aimed at ‘pleasing’ them, or building relationships, were perceived as detrimental, especially without the developmental input that they needed [42]. Thus, when addressing the HP milieu, ‘caring for’ athletes whose needs are clearly expressed and self-determined by their own long term success as performers, offers a paradox to coaches who are presented with key conceptual precursors of PS [14].

The findings from our study present clear differences between the two groups’ descriptions of their dialogue with their coaches. On the one hand, those players who were able to progress, demonstrated ‘reciprocity’ between feedback and their choice to accept it and engage with it [40,76]. Clear connections can be drawn here to two relevant concepts. Players perceived role clarity [36], that is where they fit within the wider group and the accepted roles and responsibilities that come with it, and their feedback literacy [34], which addresses how well an athlete can interact with feedback. Progression was supported when players understood where they stood in the social context and the roles and responsibilities that were expected of them. In contrast, stagnation seemed to be coupled with coach decisions about the destination of loan clubs, or (de)selection for the senior team. Similarly, the group who ‘made it’, described numerous instances where they took time to consider and process critical feedback, often relishing these types of challenges [23]. Interestingly, the coach’s capacity to care for their players in an HP environment seemed dependent on the interrelation between the capacity to engage in open dialogue with players, the creation of reciprocity in highly challenging situations, ensuring player role clarity and, finally, the nurturing of a player’s feedback literacy over time.

We hope this adds further nuance to the existing base of evidence in the coaching field, which in recent years has emphasised the interpersonal dimensions of the coaching process [77]. To be clear, this is not to suggest that abusive coaching practice is justified in any domain, not only for the obvious ethical reasons, but also for the sake of performance. Such an extreme ‘either-or’ position is unfortunately commonly expressed but not grounded in any reality of which we are aware. Where players in this sample did feel uncared for, it seemed to have a negative performance consequence. Therefore, whilst there may be a case for performance sport to use the concept of PS under very specific circumstances, we would suggest there is a need for a more mechanistic understanding of the dynamics at play. In addition, a clear need for the field to be conceptually clear, especially in the realm of TD and elite sport, where the demands of the milieu are very different to those experienced in community sport or organisational life. This is especially important if findings are to be used in applied practice, as we cannot ignore the realities of HP. In this regard, we would suggest that conceptions of care seem to present significantly more transferability, based on environmental nuance. 

The perception that the coach really cared about the player is important. Care from the perspective of the player seemed to be driven by a level of complementarity; that player and coaches converged towards the same aims [78]. Significantly, for the team sport coach, this convergence seemed to be towards the interests of the individual player rather than the organisation. In essence, there was a perception that the coach understood and cared for that player’s developmental needs and career ambitions. For some players, it was important to them that the coach understood their life beyond being a rugby player. Yet, for most players, the latter was not perceived to be an essential feature of an effective coaching experience. This is a notable finding and one that deepens our understanding of the coach in the TD and HP setting.

There are of course nuances presented by the sample given that even those players who ‘didn’t make it’ could be classified as ‘competitive elite’ [8]. Thus, it may be that the relative ‘eliteness’ of the sample has implications for desired coaching practice as all participants were, at least for a time, full time professional athletes. Thus, it is likely that their participatory aims were more geared to performance [45]. We might justifiably suggest that this may not be the case in populations with a more participatory focus. Therefore, we suggest that blanket statements designed to cover the realm of ‘coaching’ from the U5′s to the podium are clearly not appropriate but, rather, counterproductive and confusing.

#### 4.2.2. Soft Skills

On the basis that the role of the coach seemed to be so fundamental to the perceived progress of the players and that such a complex picture was presented by their reflections, as has long been identified, interpersonal skills are a core feature of coaching practice [79]. Taking this idea, colloquially, interpersonal skills are often referred to as being ‘soft skills’. On the basis of the data presented in this sample and others [30], this label would seem wholly inappropriate. Instead, the label ‘interpersonal skills’ seems far more appropriate to describe the range of different stances and approaches that were adopted and perceived to be supportive of player progress. In this particular instance, the interpersonal approaches adopted by coaches need to be matched to the nature of the HP milieu [17]. Perhaps challenging the common discourse around coaching that is suggesting that care in coaching may need to rely on ‘softer’ approaches, both harder and softer approaches were used by coaches [72]. Additionally, given the complexity of both the setting and the athlete’s needs, it is also clear that effective interpersonal skills were fundamentally intertwined with the coach’s ability to utilise a wide body of knowledge in their practice [80]. For example, understanding the role of feedback on the player and deciding an appropriate way to engage the athlete in a feedback process, requires pedagogic and interpersonal knowledge and skills, along with the technical, tactical, physical and psychological knowledge by which to begin a feedback process [69,80].

Players also saw the competence and power of the coach as a primary mediator of their messaging. That is, unless they believed that the coach was competent enough to offer them input, what they heard was rejected and added to a sense of overall frustration [34]. Much like the literature in pedagogic sciences, such as understanding impactful feedback as a process [34,81], the data presented here suggest a complex interaction between coach and athlete. The interpersonal effectiveness of the coach was underpinned by a range of other coaching knowledge and skills [71]. Indeed, the data presented here suggest that high quality coaching practice was supported by an effective coach–athlete relationship [67]. However, as specifically highlighted by a number of participants, this was not sufficient for high quality coaching.

## 5. Limitations

Given our attempt to successfully address the research aims, there are of course limitations in our approach. First, the retrospective nature of the methods employed within this study have often been criticised in research within HP and TD environments as they may offer an invalid and untrustworthy representation of athlete experiences [2,82]. Secondly, the analysis may have benefited from the triangulation of coach perspectives, if players had been matched to coaches that had been working with players during critical periods. This option was discounted given the desire for a longer term, career perspective of each individual player, each of whom had worked with large numbers of support personnel and, as such, coaches may only be able to offer a limited perspective on the experience of each individual. Given these limitations it is essential to first consider possible alternate methods of data collection in respect to the research aims and second, address the attempts to mitigate the impact of these limitations in respect to the research findings.

In considering the study design, alternative methods of data collection may have utilised more in situ ethnographic methods over prolonged periods of time within a single TD pathway [2]. Whilst this approach would have resulted in data that explore an environment in a much deeper and broader fashion, it also would result in a much smaller sample size and a failure to capture the similarities, perceptions and experiences of rugby players across different clubs and environments, and at different times, who eventually ended up at the same ultimate destination. Furthermore, to mitigate athlete data, which may be invalid or untrustworthy, the graphic timeline task successfully stimulated the player’s memory and supported the recall of particular events, perceptions of the environment and experience of the coach’s support [83]. This is a likely reason as to why the member reflection process corroborated the tabulation of our TA.

Finally, to mitigate possible survivorship bias, the inclusion/exclusion criteria was employed to explore two populations who had experienced the same overall TD system, albeit within different contexts, which resulted in different athlete perceptions. To truly address the research aims, the perspective of athletes from both populations were valued, offering an equal investigation into those who had and had not made it [47].

## 6. Applied Implications

This paper raises several challenges for the coach, coach development and broader coaching practice, perhaps reflecting a level of nuance that is very difficult to capture outside of the complexity of the context and social milieu where coaching happens. To this point, much of the literature that has considered the role of the coach has focused on the softer side of the interpersonal dimensions of the coaching relationship. In practice, this has meant that much coach development support has tended to be unable to offer coaches support on harder interpersonal approaches (e.g., role clarity, provision of challenge). We therefore believe that there is a significant opportunity for practitioners to expand the scope of their work beyond the boundaries of ‘soft skills’ and begin to consider the extent to which their interpersonal approach deploys a range of both hard and soft approaches [72]. Importantly, this is not an ‘either/or’, but, rather, should be seen as an ‘and/both’.

### 6.1. Role Clarity

We would suggest that for the individual coach, the suggestion that coaches should spend time explaining to the athlete the ‘why’ behind their coaching decisions would be a good starting point [84]. In addition, engaging in open dialogue with athletes may be the necessary starting point for effective coaching, especially if we are to ensure that care is a core feature of the coaching environment [39]. Without this open but potentially uncomfortable dialogue [85], there is a risk that athletes see the coach as a nice person who is unable to help them improve, especially in HP populations. Athletes may well need to feel psychologically unsafe, at least in the short term. A core dimension of open dialogue should be the athlete’s generation of role clarity (cf. PCDEs)—[86]. The coach has a critical role to play in minimising role ambiguity as a means of driving a player’s developmental journey, especially as they work through a variety of challenging experiences [36]. This means players knowing what their role in a group was, what a coach expected of them and what they would be judged on. This may be especially important as players navigate through a psychologically unsafe landscape.

With no exceptions, all players in this sample experienced issues with coaches and practice that they considered to be poor. This suggests important implications for player development, where TD systems may need to consider the extent to which players are prepared to work with coaches who are unable to provide optimal developmental conditions. This may be especially important in settings where player development is more distributed across a wider network of influences (e.g., [68]). How the coach optimally prepares athletes for periods where they may have to be the predominant driver of their development seems to be critical [87]. Importantly, unlike the experiences of a player in this sample, this does not mean leaving players to their own devices and calling it empowerment.

### 6.2. Safety

Perhaps controversially, we would suggest that, rather than being an implicitly desirable feature of all coaching environments, this study clearly shows that a lack of PS was an ever-present feature of the experience of the players in this sample. Indeed, in contrast to the literature in the organisational domain, a lack of safety was often regarded by individual players as a performance enhancer [88]. As a result, it seems that PS, where present, rather than being universally positive may limit development, cf. [14,15]. In essence, there is the fascinating dimension, clearly warranting further empirical attention, that so long as coupled by a level of role clarity, it may be desirable for coaches to deliberately reduce PS. At the very least, we would suggest that the uncritical attempt to apply the construct across performance sport settings without an evidence base is both unwarranted and potentially harmful to the athlete interests presented as of paramount importance. For the time being, it would at least appear wholly unwarranted for PS to be seen as an unquestionably positive element of athlete development, or in many cases whether it is actually possible for PS to be a feature of the HP milieu [14].

Whilst it appears that HP athletes will rarely experience PS as defined by Edmondson [11] or Vella et al. [13], suggestions that the HP milieu can be made psychologically safe are unlikely, and perhaps, undesirable. However, in specific conditions, a coach may choose to generate a specific set of circumstances that allows for increased perceptions of safety, either to manage the fatigue of an athlete, or where they may be struggling with the demands of the environment. This circles on the centrality of role clarity as a means of supporting performance and player development. If safety is seen on a continuum, rather than the characteristic of an environment, there may be situations where higher levels of safety may be adaptive. For example, players in this sample discussed transitioning out of international squads, back to their club, or where players were exhausted from constant judgement and competitive demands. For this reason, there are potentially transferable dimensions of the concept of the ‘safe container’, a temporally constrained block of time where increased perceptions of safety can be generated via contracting lower levels of judgment [89].

Thus, as has been the case for coaching practice for a long time, the coach has the chance to deploy strategies such as telling an athlete that they will be selected for a given period of time, regardless of their performance, or an athlete might be removed from competitive games for a block of time to make technical adjustments. By the same token, just as was the case for athletes in this sample, coaches may remove a sense of safety but take advantage of a lack of safety, offering role clarity for the athlete. For example, in a centimetres, grams, seconds (CGS) sport, an athlete will know the performance level they might need to reach to achieve a higher level of funding or selection for competition. Similarly, in team sports, an example is the time taken by the coach on an individual basis to show and articulate to the player exactly what they need to do to be selected. In all of this, it is of course worth noting that if one athlete is guaranteed selection, then another athlete is made unsafe because they are not being selected.

Much of the data presenting in this sample bears similarities to discussions of learning and performance more generally [90]. There is obviously more research to be conducted, and whilst the affective conditions for learning are generating more attention [30], there is more research to be carried out that explores the potential utility of a range of emotional experience on learning and development. Finally, in an era where coaches of all ages and stages are wrestling with a variety of athlete welfare concerns, there is an essential need for careful, granular interrogation of concepts and the realities of the HP domain. As per the data in this sample, more consideration needs to be paid to the wants and needs of HP athletes and data from the participation setting should not be universally applied. PS is not a term that should be used loosely, it is a well-established and empirically supported concept in specific organisational settings [12]. This may prevent the confusion amongst practitioners in the HP sport community, which, anecdotally at least, seems to be causing such a range of issues between athletes and coaches.

## Figures and Tables

**Table 1 sports-10-00083-t001:** Participant career status.

	Career Status at Time of Interview	International Status
Player 1 (S1)	Professional player at Premiership club	U18/U20/Senior
Player 2 (S2)	Professional player at Premiership club	U20/Senior
Player 3 (S3)	Professional player at Premiership club	U20/Senior
Player 4 (S4)	Professional player at Premiership club	U18/U20/Senior
Player 5 (S5)	Professional player at Premiership club	U18/U20/Senior
Player 6 (S6)	Professional player at Premiership club	U18/U20/Senior
Player 7 (S7)	Professional player at Premiership club	U18/U20/Senior
Player 8 (R1)	Released	U18
Player 9 (R2)	Released	U18/U20
Player 10 (R3)	Released	U18
Player 11 (R4)	Released	U18/U20
Player 12 (R5)	Released	U20
Player 13 (R6)	Released	U18/U20
Player 14 (R7)	Released	U18/20
Player 15 (R8)	Released	U20

**Table 2 sports-10-00083-t002:** Player perception of effective coaching.

	Characterised as Enabling	Raw Data Example	Characterised as Disenabling	Raw Data Example
Harder coaching approaches	High challenge and accountability to high standards	It was robust, often negative, tough love if you like, but I needed it. It was whenever I had a coach who was hard on me, but respected me as a person, that’s what I always took the most from, seeing me as more than a player (R5)	Lack of accountability or challenge from coach	(1st team coach) came in as head coach and I got along really well with him. It was brilliant, my mate is the head coach kind of thing. I had (1st team coach) who was practically like my rugby Dad, he was looking after me, putting his arm around me. On a personal level it was immense, but my performance wasn’t good, it was the time I stagnated the most probably because I felt so comfortable. (S6)
	Offering Role clarity	(International coaches) I learned so much from them, they were really personal with it. They told me exactly what they saw from me, and what I needed to do to improve and it was a really good environment to learn. (S4)	Ambiguity of role	I came off a loan and I felt as though I started to make a bit of progression. Then I got told I was meant to play in the European games by (head coach). He told me before that I was going to get picked and then for whatever reason I didn’t get picked for those games (R1)
	Robust two-way and actionable feedback	I had a conversation with (coach) about selection. I asked why another player was being picked over me and (coach) said: ‘he’s more physical in the carry and it suits the game this week’. I’d say: ‘I’ll work on being more aggressive than the carry’. He said: ‘No, you’re not the same carrier as (player). Keep working on what you’re good at. Your link play, your tips and playing out the back. You’ve got good feet you’ve got good handoff, use that to beat defenders, not just run straight through them like (player) does’. That kind of feedback is massive (S2)	Non-actionable, or inauthentic feedback	I like a coach that tells me what I’m doing well, what I’m not doing well and how I need to improve on it. Instead, at that time, all I was getting was coaches just trying to please me and build relationships. It was like ‘oh yeah, you’re doing really well’ and then realistically I just wouldn’t get any feedback. So I felt like that was a real point of stagnation. It made me angry at the time (S5)
	Building understanding through attention to detail	Having the technical knowledge is absolutely massive. With (1st team coach), I had never seen a coach who could say: ‘watch this clip from a game in 2010, it would be perfect for you’ to help you. I was sitting at home one night and he messaged me at 10 pm, he asked if I was watching the European Cup game and said: ‘someone just made a really good read, I thought you would want to see it’ (laughter). He was so passionate and wanting to help you. The tiny details have helped so much. (S7)	Lacking attention to detail	No one had ever said: ‘you’re better off carrying the ball into hands, because then you can pass, kick and run. No one cared about that, which I think, you know, looking back, someone’s job probably was to say ‘try carrying in two hands’. It’s such a fundamental part of the game to be successful, particularly in my position. You need that attention to detail and that accountability for developing. (R2)
Softer coaching approaches	Care and empathic accuracy	I was developing because (school coach) was someone who has been really important to me. I still speak to him quite frequently, he’s someone that cares about your wellbeing and cares about you as a person, but he was still pushing me, expecting better performance. (S4)	Lack of care	If coaches called you the wrong name, or your name was being missed off team sheets for training, you don’t feel great. It’s a big thing for you, but the coaches won’t think too much about it. (R3)
	Coach openness to player input	I asked (head coach) for a chat. We sat down and I said: ‘I want you to just to have faith in me. Just trust me. I’m going to work as hard as I can. We found some common ground. Usually, players keep it all on their chest. It was just a very good open conversation, now I think he’s one of the best coaches I’ve ever had. It just took time for us to find common ground. (S6)	Coaches unavailable or unapproachable	When the pressure is on, coaches turn to focus on winning every week. At that point, they become less available. I’d go 8 weeks without speaking to them. You’d end up waiting around for hours trying to chase people. It got to the point where there was no dialogue, it was just a loan sheet. It was no one’s job to check on us. I remember them saying: ‘own your own development’. They never saw it as their job to help you. (R4)

## Data Availability

Not applicable.

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
