# Peer review of "Tough Love—Impactful, Caring Coaching in Psychologically Unsafe Environments"

_sports, 2022, doi:10.3390/sports10060083_

Round 1
Reviewer 1 Report
Well organized and provided significant issues regarding coaching science from the persectives of sports psychologist.
Author Response
We thank reviewer 1 for this comment and overall review.
Reviewer 2 Report
This paper is well written, executed, and includes in-depth discussions on the topic.
Please spell out TD in line 78 when it’s first introduced. Also, I see many abbreviations, HP, PS, and sometimes it’s just hard for readers to know that those are as reading the paper (or when some skip the first part of the paper) and I think the words are not really long, so I’s suggest using the full words throughout the paper.
In addition the table, if the authors visualize the findings using a figure, that would be helpful.
Author Response
We thank reviewer 2 for these helpful comments. Please see below a record of changes:
Please spell out TD in line 78 when it’s first introduced.
Thank you, this is now changed
Also, I see many abbreviations, HP, PS, and sometimes it’s just hard for readers to know that those are as reading the paper (or when some skip the first part of the paper) and I think the words are not really long, so I’s suggest using the full words throughout the paper.
Thank you, we have reduced the number of abbreviations by removing HP and swapping with high performance throughout (these changes are in red in the manuscript).
In addition the table, if the authors visualize the findings using a figure, that would be helpful.
We thank the reviewer for this suggestion, though we are unsure what type of figure is being suggested. This would be a little unusual for this type of research. If this is required, please can you provide a little more information?
Reviewer 3 Report
This is a well developed and well argued paper, making an important contribution to our understanding of relationships between coaching, psychological safety and performance, in a specific elite sport context. The authors engage well with the literature, identifying gaps that need to be addressed. They have done very well in collecting a rich data set of interviews, which provides the basis for the contribution of the paper. The identification of the importance of context and of relationships is particularly welcome. Well done.
Some minor comments:
-It would help to define 'elite sport' directly, and early on in the paper;
-The term 'TD' (line 78) is introduced without being defined. I assume it means 'talent development', based on the key words. Similarly, TID is introduced later on, without noting what it stands for;
-Line 192, it is noted that, importantly, interviews occurred within a certain time period. Another sentence or phrase indicating why this was important would help;
-The graphic timelines are very interesting as a data form. If it was possible, I would encourage the authors to include an example of such a timeline in the paper;
-Table 2 has a lot of important material, but is introduced rather briefly. Another sentence or two at line 437 explaining the key insights of the Table would be beneficial to the reader.
The following comments are focused on areas for future research directions, action is not required on them for this paper. They are thoughts I had as I was reading the paper that the authors may like to consider if they develop this research trajectory further.
-The focus on coaches is clearly appropriate for this paper. In future research, some more analysis of the roles of different coaches within a club would be interesting to consider eg head coach contrasted with a line coach. Similarly, elite sport teams also have many other people working in this area, including psychologists. It would be interesting to consider the interactions and relationships between these different actors in this context.
-As noted by the authors, both in the context of the literature review and in the context of their findings, the concept of psychological safety is open to questioning and debate. The authors begin to do this, but there is now an important opportunity for them to extend this analysis further and to consider more directly conceptual alternatives or extensions to the concept.
Author Response
We thank reviewer 3 for these very helpful comments. Our responses are in red below:
Some minor comments:
-It would help to define 'elite sport' directly, and early on in the paper;
-The term 'TD' (line 78) is introduced without being defined. I assume it means 'talent development', based on the key words.
Thank you, this is now changed
Similarly, TID is introduced later on, without noting what it stands for;
This is also now changed to TD, which is more appropriate
-Line 192, it is noted that, importantly, interviews occurred within a certain time period. Another sentence or phrase indicating why this was important would help;
Thank you, this comment is to enable the reader to ground themselves in understanding the perspective of the player. Addition to L194 as follows:
"Importantly, for perspective on the career status of the player"
-The graphic timelines are very interesting as a data form. If it was possible, I would encourage the authors to include an example of such a timeline in the paper;
Thank you for this suggestion and agree that they are interesting as a data form. However, for this reason we decided against their addition because we feel they would detract from the specific RQs being answered in this piece of research. In essence, because they were used as a tool to support recall they do not provide insight on either extent of psychological safety, or enabling coaching practice.
-Table 2 has a lot of important material, but is introduced rather briefly. Another sentence or two at line 437 explaining the key insights of the Table would be beneficial to the reader.
Please see addition to the manuscript at L440:
The perspectives of the players suggest a non-dichotomous perspective on effective coaching. Players strongly believed that effectiveness was the result of both ‘harder’ and ‘softer’ approaches to coaching practice. In short, players strongly desired highly competent coaches who presented them with a clear direction and held them to high standards, whilst also caring for and about them.
The following comments are focused on areas for future research directions, action is not required on them for this paper. They are thoughts I had as I was reading the paper that the authors may like to consider if they develop this research trajectory further.
-The focus on coaches is clearly appropriate for this paper. In future research, some more analysis of the roles of different coaches within a club would be interesting to consider eg head coach contrasted with a line coach. Similarly, elite sport teams also have many other people working in this area, including psychologists. It would be interesting to consider the interactions and relationships between these different actors in this context.
Thank you, this is a good idea, it certainly raises a few interesting questions
-As noted by the authors, both in the context of the literature review and in the context of their findings, the concept of psychological safety is open to questioning and debate. The authors begin to do this, but there is now an important opportunity for them to extend this analysis further and to consider more directly conceptual alternatives or extensions to the concept.
Thank you for this, we agree and hope to extend the research in the future. We also hope suggestions in the manuscript are worthy of practical application.